# SEEP-CI: A Structured Economic Evaluation Process for Complex Health System Interventions

**DOI:** 10.3390/ijerph17186780

**Published:** 2020-09-17

**Authors:** Jason Madan, Meghan Bruce Kumar, Miriam Taegtmeyer, Edwine Barasa, Swaran Preet Singh

**Affiliations:** 1Warwick Medical School, University of Warwick, Coventry CV4 7AL, UK; S.P.Singh@warwick.ac.uk; 2Department of International Public Health, Liverpool School of Tropical Medicine, Pembroke Place, Liverpool L3 5QA, UK; meghan.kumar@lshtm.ac.uk (M.B.K.); miriam.taegtmeyer@lstmed.ac.uk (M.T.); 3MARCH Centre, London School of Hygiene and Tropical Medicine, London WC1E 7HT, UK; 4Health Economics Research Unit, KEMRI-Wellcome Trust Research Programme, Nairobi 00100, Kenya; ebarasa@kemri-wellcome.org; 5Centre for Tropical Medicine and Global Health, Nuffield Department of Medicine, University of Oxford, Oxford OX3 7LG, UK; 6Coventry and Warwickshire Partnership NHS Trust, Coventry CV6 6NY, UK

**Keywords:** complex intervention, economic evaluation, health system interventions

## Abstract

The economic evaluation of health system interventions is challenging, and methods guidance on how to respond to these challenges is lacking. The REACHOUT consortium developed and evaluated complex interventions for community health program quality improvement in six countries in Africa and Asia. Reflecting on the challenges we faced in conducting an economic evaluation alongside REACHOUT, we developed a Structured Economic Evaluation Process for Complex Health System Interventions (SEEP-CI). The SEEP-CI aims to establish the threshold effect size that would justify investment in a complex intervention, and provide an assessment to a decision-maker of how likely it is that the intervention can achieve this impact. We illustrate how the SEEP-CI could have been applied to REACHOUT to identify outcomes where the intervention might have impact and causal mechanisms, through which that impact might occur, guide data collection by focusing on proximal outcomes most likely to illustrate the effectiveness of the intervention, identify the size of health gain required to justify investment in the intervention, and indicate the assumptions required to accept that such health gains are credible. Further research is required to determine the feasibility and acceptability of the SEEP-CI, and the contexts in which it could be used.

## 1. Introduction

The Lancet Global Health Commission on High Quality Health Systems in the Sustainable Development Goal era highlights the enormous scope for improving health through investment in health systems, and the need for such investment to be efficient [1]. Economic evaluations of complex health system interventions (e.g., providing training, creating a clinic or health facility, or increasing staff numbers) are required to support this goal. However, in our experience, economic evaluations are disproportionately skewed toward evaluating health technologies (e.g., a medicine, a device, or a clinical management approach) rather than health system interventions. This bias towards economic evaluation of health technologies over health system interventions can be seen in the UK, for example, where the National Institute for Health and Care Excellence has a process for health technology appraisals [2], but does not have an equivalent for health system interventions.

It is likely that insufficient economic evaluations of health system interventions exist because they are challenging to conduct, rather than because they are not needed. While guidance exists on the development and evaluation of RCTs for complex interventions [3] and on the evaluation of complex public health interventions [4], such guidance pays little attention to economic evaluation [5]. There are no such guidelines for economic evaluation of health system interventions, in contrast to health technologies [6]. There is growing interest in developing and consolidating methods for the economic evaluation of health system interventions, reflected in several methodological and editorial publications in the recent literature [7,8,9].

We present here a case study involving a complex health system intervention from the REACHOUT consortium [10], implementing quality improvement (QI) approaches with community health workers (CHWs) in six countries. We use this case study to illustrate the challenges that can arise in determining whether such interventions are cost-effective, and outline a Structured Economic Evaluation Process for Complex Health System Interventions (SEEP-CI), which we developed in response to these challenges. We use the case study to show how the SEEP-CI combines robust empirical evidence with analysis that makes explicit what decision-makers would need to assume to justify implementing the intervention. We discuss how SEEP-CI could provide an efficient approach to generating useful information for decision-makers, particularly in lower- and middle-income countries (LMICs).

## 2. Materials and Methods—Development and Outline of the SEEP-CI

### 2.1. Description of Case Study

Community health programs (CHPs) in LMICs rely heavily on CHWs given the shortages of professional health workers. CHWs provide services such as health education, counselling, screening, diagnostics, and treatment. REACHOUT was a program of research motivated by the belief that investing in quality improvement (QI) for CHWs, and CHPs more generally, could lead to substantial and cost-effective health gains across the many disease areas that they address, by improving their effectiveness in providing the services listed above. REACHOUT developed a suite of QI interventions and explored their impact in six countries in Africa and Asia. CHW supervisors were trained in supportive supervision techniques. Local community QI teams consisting of facility and community stakeholders were trained in QI concepts and supported in using community health data to identify problems and develop and implement appropriate solutions. The teams were not required to work on any specific health area or issue due to differences in context and the local nature of community health, though most dealt with broad areas of maternal and child health. Further details of the specifics of the intervention can be found elsewhere [10,11].

### 2.2. Description of Challenges for the Economic Evaluation of Complex Health System Interventions Such as REACHOUT

Health system interventions, by their nature, are inevitably complex. Complex interventions have characteristics that create challenges for evaluation. Guidance on evaluating complex interventions highlights these challenges, including ‘the difficulty of standardising the design and delivery of the interventions, their sensitivity to features of the local context, nd the length and complexity of the causal chains linking intervention with outcome’ [12]. There are specific challenges that arise when the goal is an economic evaluation of a complex health system intervention, as Barasa et al. demonstrate for the package of care interventions [13]. The REACHOUT suite of interventions clearly featured these challenges, and we illustrate below how this created difficulty for an economic evaluation of these interventions.

#### 2.2.1. Dispersed Impacts

CHWs influence numerous process and health outcomes, depending on the organization of the health system they interact with, and the prevalent health issues in the populations that they serve, as well as one-off health challenges, such as pandemics, that can be unpredictable. This made it difficult to design data collection on the benefits of the REACHOUT QI interventions. A further concern, similar to that identified for patient safety interventions [14], was that the impact of QI interventions might comprise a large number of small benefits across diverse patient groups, making it difficult to establish whether the overall impact was large enough to make interventions cost-effective.

#### 2.2.2. Non-Standardized Interventions

REACHOUT involved a suite of complex interventions that were iterated over time to improve their effectiveness. While there was an overarching consistency across their implementation from the study design, those who implemented the interventions inevitably did so in ways that could not be fully specified in the protocol, and varied between countries. For example, local QI teams were free to develop and implement changes that they saw as priorities, and these initiatives inevitably varied across countries, as they were operating in different contexts, and faced different resource constraints. Some QI teams focused on training, for example, while others identified routine data reporting as a priority.

#### 2.2.3. Complex Causal Relationship between Intervention and Outcome

REACHOUT interventions were targeted at improving outcomes such as referral rates, or how often CHWs identified red flag symptoms correctly. These were intended to lead to health gains, for example, through earlier presentation, leading to reduced risk of complications and better outcomes, but the complex causal chain between action and health allowed for substantial potential for confounding, through factors such as disease outbreaks, economic hardship, or political change. Essentially, the arms-length relationship between intervention and health effect created a ‘signal-vs.-noise’ problem.

#### 2.2.4. Impact Conditional on Health System Functioning

The impact of Community Health QI depended on the health system in which CHWs worked. REACHOUT might have succeeded in improving CHW skills and motivation, and this might have led to higher numbers of appropriate referrals, but the health system might then have lacked the ability or resources needed to provide adequate care for those referred, due to issues such as lack of medicines, unwelcoming staff at the facility of reference, or numerous other demand or supply constraints that apply in LMIC settings [15]. In that case, the intervention would appear to have failed, despite its achievements.

## 3. Results—Development of the SEEP-CI and Illustrative Application to an Evaluation of Embedding QI in Community Health Programs

### 3.1. Development of the SEEP-CI

The conventional approach to gathering the evidence required would be to conduct a randomized controlled trial (RCT), in which the primary outcome(s) would be relevant to the decision being informed. The combined impact of the challenges described above is that, for such a study to be definitive, the sample size would need to be unfeasibly large. We propose an alternative approach that aims not to establish a precise estimate of the costs and benefits associated with the health system intervention, but instead asks ‘how plausible is it that the intervention is worth investing in’? This approach blends the strengths of RCT design with a structure drawing on formal decision theory [16] that allows policymakers to exercise judgement in interpreting available evidence to determine investment priorities.

The SEEP-CI evolved over several years during the REACHOUT project. Its development was informed by discussions with consortium researchers and other stakeholders, such as national and district level health workers and officials, in each of the participating countries. That process involved identifying the challenges described above and eliciting qualitative data from policymakers on what they perceived to be the uses and shortcomings of conventional economic evaluations [17]. While those discussions were helpful in identifying the shortcomings of conventional economic evaluation approaches for decision-making, the SEEP-CI was developed by the authors with no formal input from external experts, thus no formal procedure for reaching consensus about the steps involved (e.g., the DELPHI method [18]) was conducted or required, and the steps of the SEEP-CI, described below, solely reflect the views of the authors.

### 3.2. Description of the SEEP-CI

The SEEP-CI comprises the following six steps.

#### 3.2.1. Identify and Categorize Outcomes of the Intervention

We use the term ‘outcome’ to mean anything that might change, directly or indirectly, if an intervention were implemented. For health technologies, key health outcomes can often be easily identified. Health system interventions are likely to have more wide-ranging and diverse outcomes, given their complexity and impact on multiple clinical and health system pathways, and process outcomes can be as informative as health outcomes. The first step in the SEEP-CI is therefore to identify, as comprehensively as possible, all the outcomes that might be affected by the intervention. To aid this process, we suggest categorizing outcomes along two dimensions:

Intrinsic vs. Instrumental—Intrinsic outcomes are those that stakeholders value for their own sake, such as mortality or cases of disease averted. Instrumental outcomes are those that are not valued in their own right, but generate value by influencing intrinsic outcomes, such as blood pressure levels or staffing ratios.

Proximal Vs. Distal—Proximal outcomes are those directly affected by an intervention, such as the number of staff who have received training. Distal outcomes are those where there is a multi-link causal chain linking the outcome to the intervention, such as the mortality rate for a given condition in a population of interest.

Whether an outcome is instrumental or intrinsic may depend on the perspective of the stakeholder (and some may be both), and the distinction between proximal and distal is one of degree. However, the purpose of this classification is to deepen understanding of potential mechanisms through which the intervention might effect change, and recognize, as broadly as possible, where the intervention might create value. Therefore, the process of identifying and categorizing outcomes, and the discussions involved in doing so, are of greater importance than the category to which outcomes are assigned. Moreover, the relevance of an outcome will depend on the decision perspective. Our examples are health-related, but outcomes beyond health should also be considered at this step if they are likely to be of importance to the decision-maker.

#### 3.2.2. Develop a Conceptual Model Outlining the Causal Relationship between the Intervention and the Outcomes Identified in Step One

A conceptual model is a representation of a system that describes its elements and the causal relationships between them [19]. The aim of developing such a model is to identify the mechanisms through which the intervention might generate impact. With health system interventions, there is likely to be a complex causal network linking outcomes, with feedback loops reinforcing or attenuating impacts. The aim of constructing the conceptual model is to capture, share and refine an understanding of what the effects of the intervention are, how they ripple through the complex system in which they act, how feedback loops might reinforce or counter effects, and where potential for confounding exists. To construct such a model, it will be necessary to:(i)understand what the proximal goal of the intervention is, such as delivering a training package to a specific cadre of health workers(ii)specify the relationship between this goal and health care delivery e.g., better trained workers might miss fewer cases where referral to specialists was required(iii)determine the consequences of changing health delivery e.g., whether more appropriate referrals might improve survival rates(iv)identify potential feedback loops e.g., improved survival might increase patient willingness to engage with the health system, leading to more opportunities for health workers to learn through experience

Identifying all these causal links will require input from all stakeholders—health workers, managerial staff, patient groups—and formal processes exist to facilitate this [20]. Going through this process of identification is likely to suggest additional outcomes not included in Step One. Therefore, Steps One and Two are best conducted iteratively. Identifying outcomes will prompt stakeholders to think of plausible mechanisms linking them and identifying mechanisms will lead to further outcomes of interest being identified.

#### 3.2.3. Identify Key Intrinsic Outcomes that Will Drive the Adoption Decision

Step One will have identified numerous intrinsic outcomes across multiple clinical areas. We recommend that the study team select a subset of these outcomes for evaluation, perhaps even as few as one or two. The intrinsic outcomes selected should be ‘decision drivers’ i.e., those thought to be most influential in determining whether an intervention is cost-effective. Decision drivers will be outcomes where the intervention is likely to have the greatest total impact, and the evidence supporting that impact is strongest. The size of achievable impact will be greater for commonly occurring events with severe health consequences and/or high costs. The actual impact resulting from the intervention can be estimated more precisely and convincingly when the causal chain linking intervention to impact is short, and the links in that chain are strongly supported by evidence and expert knowledge. This reduces the potential for confounding and the likely uncertainty around recommendations for decision-makers. The rationale for this step is that it will usually not be feasible to capture the impact of the intervention across all the outcomes identified in Step One, so, following the criteria described here will ensure the efficient use of available study resources.

#### 3.2.4. Estimate the Cost of Delivering the Intervention, and the Threshold Change in the Key Intrinsic Outcome(s) Required for the Intervention to Be Cost-Effective

It will usually be possible to estimate and cost the resources required to implement an intervention, based on the study protocol. If the decision-maker’s willingness to pay per unit gain in an intrinsic outcome can be established, the change in that outcome required to justify the investment can be calculated. For example, if a national training program for medical officers would cost 0.1 million USD, and the national willingness-to-pay threshold is 1000 USD per DALY, then the training program would need to result in 100 DALYs averted to justify investing in it.

The reason for focusing on the direct costs of implementing an intervention is that they are key to understanding impact on constrained budgets and resources, and that they can be calculated without requiring substantive assumptions about the knock-on impact on healthcare resource use. If these knock-on impacts are important to the evaluation, it might be appropriate to expand the threshold analysis to include them. For example, if the training program described above would require additional investment in testing and treatment resources to yield health gains, then the target for DALYs averted would rise by 1 DALY for each 1000 USD of investment required.

#### 3.2.5. Identify Proximal Instrumental Outcomes that are Likely to Cause Changes in the Key Intrinsic Outcome(s) Identified in Step Three, and Estimate the Effectiveness of the Intervention in Terms of These Proximal Outcomes

The intrinsic outcome(s) selected in Step Three will have been chosen with the simplicity and plausibility of the causal mechanism between outcome(s) and intervention in mind. Nevertheless, that causal chain is likely to be indirect, making attribution difficult. What can be established is how far the intervention is able to affect the proximal outcomes that it is designed to affect. The RCT is the gold standard study design and should be followed where possible to provide decision-makers with credible information about the effectiveness of the intervention. We propose that the primary outcome around which such a study is designed be chosen, based on the insights from Step Two, to be proximal, instrumental, and on the causal chain to the intrinsic outcome identified in Step Three. For example, if the intervention is a training program in ante natal care, the primary outcome for a study evaluating the intervention should be the numbers attending training and/or the change in their knowledge of red flags and appropriate referral routes, rather than neonatal outcomes. This does not preclude collecting data on a distal intrinsic outcome such as neonatal health, but it protects against the considerable risk of the study, failing to show a significant attributable impact of the intervention on the distal intrinsic outcome, for all the reasons explored in Section 2. In this case, it should still be possible to establish that the intervention has had an impact, albeit on the proximal instrumental outcome. The resulting impact on the distal outcome of interest can then be inferred through analytical approaches, then combine study results, the literature, and expert opinion, as per the next step.

#### 3.2.6. Assess the Strength of the Case for Investing in the Intervention

The previous steps will provide decision-makers with robust estimates of the investment required to implement a health system intervention, the size of the proximal impact it will have, and the minimum health gains that this proximal impact should generate for investment to be worthwhile. The remaining question is ‘how likely is it that the proximal impact generated by the intervention will lead to greater health gains than this minimum’? This question will require judgement from the decision-maker, as all the challenges described earlier apply here. However, there are analyses that can be provided to the decision-maker to support their assessment. Decision-analytic modelling, drawing on the conceptual model developed in Step Two, can illustrate the assumptions that need to be made to arrive at the required threshold health gain from the proximal effectiveness of the intervention. Evidence to support such assumptions can be sourced from the literature, or from expert opinion. Considerable uncertainty may still remain around the level of health gains that the intervention might generate for a given decision-maker, but the structured approach we described will provide them with the support needed to decide whether the existing evidence is strong enough to justify investment in a health system intervention, and, if not, where the key information gaps are.

Figure 1 sets out the steps involved in the SEEP-CI. If decision-makers take a view that the findings from the SEEP-CI are not sufficiently convincing for them to invest in the health system intervention, this may be because the selection of outcomes at Step Three was too narrow. The conceptual model from Step Two can be used to elicit decision-maker views on whether additional intrinsic outcomes should be added to the analysis and, if so, which ones. This can then be included in a revised Step Six analysis, for example, by additional decision-analytic modelling, to explore the sensitivity of cost-effectiveness to including additional benefits.

### 3.3. Application of the SEEP-CI to Evaluation of Embedding QI in Community Health Programmes

The SEEP-CI was developed from evaluative research carried out during REACHOUT, rather than the other way around. However, we can still illustrate how we might have structured that research if an economic evaluation of REACHOUT had been based on the SEEP-CI.

#### 3.3.1. Identify and Categorize Outcomes

Annual meetings held to bring together members of the consortium gave us opportunities for an in-depth discussion about what might change because of the intervention. We then invited consortium members to apply the intrinsic vs. instrumental and proximal vs. distal categorization described above. Table 1 gives some of the results of this process. Consortium members were able to identify several outcomes where the impact of community-level QI might be observed. Their wide-ranging role in community health means that a range of specific health outcomes (e.g., maternal mortality, diarrhea-related deaths) could be included as distal intrinsic outcomes, and that a generic health measure (e.g., DALYs) might also be appropriate. Country-level research teams identified several measures that would be proximally affected by REACHOUT interventions, most of which were instrumental. The investment required to conduct training and hold QI meetings was an obvious intrinsic proximal outcome. Additionally, it was thought plausible that staff wellbeing might directly improve through the interventions.

#### 3.3.2. Develop a Conceptual Model of Causal Links between Outcomes

REACHOUT included several linked but discrete interventions. In Figure 2, we present a conceptual model for one of them—training supervisors in supportive approaches to CHWs. The conceptual model came out of a guided workshop held in Nairobi, Kenya in April 2018, attended by REACHOUT health systems researchers from four participating countries. At the workshop, country teams developed conceptual models based on their understanding of the role of CHWs in their health systems. We integrated these models into the version presented in Figure 2, drawing on additional insights from stakeholders. The combined model illustrates the complex causal network propagating the proximal impacts of supportive supervisor training (more supportive and more frequent supervision). It includes feedback loops such as increased motivation, leading to improved quality of care, resulting in greater willingness by the community to use CHWs, further increasing their motivation.

#### 3.3.3. Identify Key Intrinsic Outcomes

During discussions on outcomes (Step One), consortium members reflected on where the impact of improving CTC provider quality through supervisor training might be greatest. Several intrinsic outcomes were proposed, depending on local context, such as avoiding complications of malaria or timely identification of infant diarrhea severe enough to require treatment. However, infant mortality was agreed to be a measure where community-level QI could yield important benefits across all participating countries, and the mechanism linking CHW supervision to outcomes was perceived to be credible and relatively direct.

#### 3.3.4. Estimate the Costs of Delivering the Intervention, and the Threshold Change Required for the Intervention to Be Cost-Effective

Given the challenges described above, we were unable to robustly estimate effects, cost-effectiveness ratios, or net monetary benefit for REACHOUT. However, we were able to conduct a detailed, country-specific exercise in costing the interventions evaluated, which has been reported elsewhere [11]. The key parameter from that costing exercise was the cost per CHW of QI. We found that the annualized cost of QI per CHW ranged from 62 USD (Mozambique) to 254 USD (Ethiopia). A full assessment of the level of benefit required to justify this investment was beyond the scope of the REACHOUT study. For illustrative purposes, assuming a marginal effectiveness of healthcare investment of 1786 USD per DALY, which was the upper threshold estimated by Woods et al. for Indonesia [21], the target DALYs gained per CTC provider would need to be 0.03 for an investment of 62 USD, or 0.14 for an investment of 254 USD.

#### 3.3.5. Identify the Key Proximal Instrumental Outcomes for the Intervention, Estimate the Effectiveness of the Intervention on these Outcomes, and Assess the Strength of the Case for Investing in the Intervention

Given that the approach that we present was developed after the design of the REACHOUT project, it was not possible to fully carry out the work required for these steps within REACHOUT. However, we can illustrate what work would be required, and how it might then be used. We base this illustration on a published economic evaluation of a community-based intervention to improve birth outcomes in rural Nepal [22]. Borghi et al. estimated that their intervention would lead to a 1% absolute reduction in infant mortality, resulting in 921 life years gained per 100,000 people covered by CHWs. We might assume that the impact of community-level QI would be mediated through increased ability to deliver interventions, such as that developed by Borghi et al. The costing conducted as part of REACHOUT found that the cost of community-level QI was between 1000 and 5000 USD per 100,000 people covered. Therefore, if CTC provider QI enhanced the effectiveness of the Borghi et al. intervention by 5%, this would result in 0.05 × 921 = 46 additional life years, giving a cost per life year gained of 108 USD. If the effectiveness were increased by 50%, the additional life years gained would rise to 460, and the cost per life year gained fall to 11 USD. For any given willingness-to-pay to gain a life-year, the target level by which QI enhanced the intervention could thus be calculated. REACHOUT documented how the CTC provider QI interventions impacted proximal measures such as motivation and QI activity, and this evidence could be provided to the decision-maker to help them assess how likely it was that the target gain in effectiveness would be met.

## 4. Discussion

The aim of the SEEP-CI is not to determine the impact or cost-effectiveness of a complex intervention as completely, precisely, and robustly as possible. Instead, it seeks to establish the threshold effect size that would justify investment in a complex intervention, and provide an assessment to a decision-maker of how likely it is that the intervention can achieve this impact. This contrasts with conventional evidence-based medicine perspectives that favor gold-standard evidence, such as RCTs. Our concern is that it is inherently more challenging to provide such evidence for health system interventions than health technologies. If the requirement for gold-standard evidence is applied inflexibility, this will lead to underinvestment in health system interventions. Nevertheless, decision-makers still need rational, evidence-based analysis to support such investment.

The SEEP-CI developed out of the difficulties that arose in attempting an economic evaluation alongside REACHOUT. REACHOUT was motivated by the observation that CHWs are an integral part of health care delivery in many LMICs, but there was limited investment in their training and supervision, or other QI measures for community health. This led to a hypothesis that investing in such measures could lead to substantial health gains at reasonable cost. While it was feasible to assess costs, the health gains were much more challenging to capture precisely. The reasons, presented here, are commonly found in health system interventions, and largely arise because we are intervening at a system rather than a patient level. As a result, impacts are diffused across multiple clinical conditions, with many causal steps between intervention and intrinsic outcome. Our view was that a case could be made in support of the hypothesis, but not a conclusive one, and that decision-makers needed to be presented with this case so that they could make an informed judgement. The SEEP-CI provides decision-makers with this case.

The challenges that arise when evaluating complex interventions, and the problems this creates for conducting RCTs, are well understood [23]. These challenges have led to the development of approaches such as realist evaluation, which explicitly considers the mechanisms and context in which an intervention acts [24], and has proven to be valuable for health systems research [25]. The SEEP-CI is naturally complementary with realist evaluation, sharing its focus on going beyond empiricism to consider mechanisms. It does differ from realist methodology in taking a decision-theoretic perspective on research design. The key consequence to this perspective is the approach to uncertainty. Reframing the question from ‘Does the intervention work’ to ‘Should a funder invest in it?’ leads to the recognition that there are costs involved in reducing uncertainty, so that it may be optimal to invest in an intervention, even if the evidence supporting it has limitations. We also draw on a body of decision theory known as ‘Soft OR’ [26]. Soft OR provides methods for incorporating into decision analysis aspects of a decision problem that are subjective and amenable to multiple perspectives. The use of conceptual models, the integration of a broad range of stakeholders and their perspectives into the analysis, and the iterative application of steps to allow feedback from later steps to revise work from earlier ones, are all aspects of the SEEP-CI that draw on insights from soft OR approaches such as problem structuring methods [27] and soft systems methodology [28].

The SEEP-CI involves several potentially subjective choices, such as the selection of intrinsic outcomes, the identification of links in the causal network, and the assessment of the strength of the cause-effect relationship between proximal and distal outcomes. We overcome this possibility of subjectivity in two ways. Firstly, we highlight the importance of establishing, objectively and empirically, the effectiveness of the intervention in terms of key proximal outcomes. This should be done using the most robust study design possible, ideally via an RCT, to provide the strongest evidence possible of the proximal impact of the intervention. Secondly, the SEEP-CI separates what is known from what is assumed, and explicitly sets out, via threshold analyses, the minimum that must be assumed to justify investing in the intervention. The credibility of the analysis will be strengthened by involving as wide a range of stakeholders as possible in each of the steps of the SEEP-CI. It is the responsibility and the privilege of the decision-maker to determine whether the investment case generated by the SEEP-CI is strong enough.

The SEEP-CI described here is an exploratory attempt to formalize an approach to the economic evaluation of complex health system interventions. It is intended to enhance, rather than replace, existing guidelines for the conduct and reporting of economic evaluations, such as those provided by the ISPOR CHEERS Taskforce [29]. Such guidelines set out core principles for economic evaluation that apply equally to health system interventions as health technologies, such as clearly defining the intervention and the decision population, identifying an appropriate time horizon and discount rates, providing sources for costs, and so on. We expect the SEEP-CI to be a helpful enhancement to existing guidance for the economic evaluation of complex health system interventions, based on strengths such as its focus on measuring proximal outcomes, creating and validating conceptual models, and threshold analyses. We anticipate that it will be particularly valuable in LMIC settings, given the logistical and resource challenges involved in conducting primary research in such settings. However, we accept that there are limitations to the SEEP-CI currently that would need to be addressed before it can be credibly used. We have yet to test the SEEP-CI by using it from the start of the study. Moreover, the SEEP-CI is our proposed solution to the challenges of evaluating health system interventions, rather than reflecting the input of multiple experts synthesized through a formal consensus process, as was involved in developing guidelines such as CHEERS. Further research validating the SEEP-CI, exploring its feasibility, acceptability to decision-makers, and ability to generate credible and helpful results, is required. A formal task force would be particularly suitable to developing the SEEP-CI further. We believe that this would be worthwhile if the approach proves to be valuable, and leads to greater and more efficient investment in health system interventions, then the potential health gains are substantial. Furthermore, while the approach we set out here focuses on allocative efficiency, decision-makers are often interested in addressing health inequity as well as maximizing health gains. Further development of the SEEP-CI could involve extending the approach to reflect this need, for example by including subgroup analyses, or ensuring equity-based outcomes are formally included.

## 5. Conclusions

The SEEP-CI can assist decision-makers by separating out what is known from what must be assumed, and by providing guidance about what criteria must be met to make investment in a complex intervention worthwhile. The credibility of the analysis can be enhanced by explicitly involving as broad a range of stakeholders as possible in each step, including the decision-maker. The SEEP-CI can be used to highlight exactly which assumptions are most disputed, and assess how sensitive the decision would be to these assumptions. It also allows decision-makers to take a conservative approach by ‘raising the bar’ i.e., by raising the cost-effectiveness threshold applied to reflect the lesser quality of the evidence. We hope that, through these measures, decision-makers will feel more comfortable making decisions around investing in complex health system interventions, and the overall efficiency of health systems will improve substantially as a result.

## Figures and Tables

**Figure 1 ijerph-17-06780-f001:**
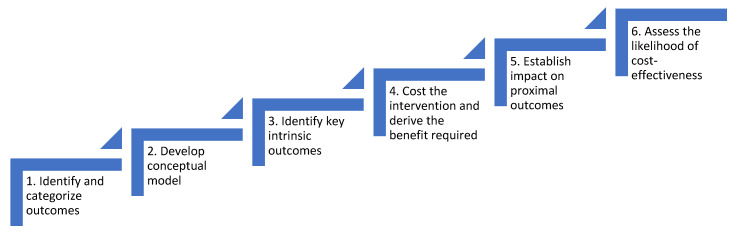
Depiction of the Structured Economic Evaluation Process for Complex Health System Interventions (SEEP-CI).

**Figure 2 ijerph-17-06780-f002:**
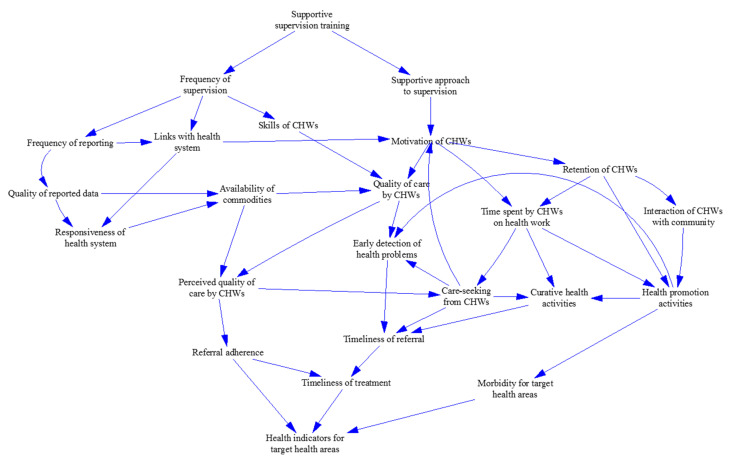
Conceptual model illustrating mechanisms, through which training of those supervising CHWs might result in health gains.

**Table 1 ijerph-17-06780-t001:** Outcomes impacted by REACHOUT, categorized as instrumental vs. intrinsic and proximal vs. distal.

	PROXIMAL	DISTAL
INSTRUMENTAL	Supervisory support for CTC providers CTC provider motivation Number of quality improvement (QI) meetings Number of QI initiatives proposed Number of QI teams trained in the global curriculum	Utilization of health clinics Community satisfaction with CTC providers CTC provider turnover Uptake of vaccinations
INTRINSIC	Costs of training Staff wellbeing	Maternal mortality DALYs Deaths due to diarrhea

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
