# Peer review of "SEEP-CI: A Structured Economic Evaluation Process for Complex Health System Interventions"

_ijerph, 2020, doi:10.3390/ijerph17186780_

Round 1

Reviewer 1 Report

The authors developed and suggested a new guideline (SEEP-CI) for conducting economic evaluation about complex health system interventions and showed an adjused example(REACHOUT).

This is an interesting report about suggesting a guideline of economic evaluation mentioned above and showed a practical example following the guideline. However, there are some questions related to the research background for developing this new guideline, and might be some problems which need to be addressed.

Compulsory Revision

Material and method

     Please describe the research methods clearly.

  1. The author should describe the research methods about the reaching agreement process of SEEP-CI contents. (if this process had been conducted and published separately, please cite the related reference.)
  2. Accordingly, the author should describe the research methods of adjusting example(REACHOUT) separately, and so,  "2.3 Description of the SEEP-CI" should be moved to the "Results".
  3. Moreover, the author should mentioned the analytic methods or process for identyfing critical values from the prior research results of REACHOUT program (related to 3.4)

Results.

     The author should present the results of economic evaluation about REACHOUT as tables and figures (costs, outcomes, ICER, NMB mentioned in 3.4)

Discussion.

     Even though there are might be imitations for conducting economic evaluation of complex health system interventions based on existing protocols and guidelines, the author should clearly show the superiority of proposed guideline by comparing every contents with existing guidelines.

Author Response

We thank the reviewer for their efforts and their helpful and insightful feedback. We have considered their comments carefully and made amendments to our manuscript where appropriate. We give details of our responses and amendments below, in italics.

Reviewer 1

The authors developed and suggested a new guideline (SEEP-CI) for conducting economic evaluation about complex health system interventions and showed an adjused example(REACHOUT).

This is an interesting report about suggesting a guideline of economic evaluation mentioned above and showed a practical example following the guideline. However, there are some questions related to the research background for developing this new guideline, and might be some problems which need to be addressed.

Authors’ response: We thank the reviewer for raising these questions. In response, we would seek to emphasise that the SEEP-CI was developed within the research team and by the authors. Although a number of stakeholders provided insights around the mechanisms through which CHW QI could provide benefits, and helped identify the challenges involved in conducting an economic evaluation of CHW QI, they were not involved in developing the SEEP-CI itself. Therefore, no formal process to synthesise views on what steps should be included in the SEEP-CI and reach agreement on its final structure was adopted or required. We have amended the manuscript to clarify this, and give details of these amendments below.

Compulsory Revision

Material and method

     Please describe the research methods clearly.

    The author should describe the research methods about the reaching agreement process of SEEP-CI contents. (if this process had been conducted and published separately, please cite the related reference.)

Authors’ response – The SEEP-CI steps were developed collaboratively by the authors, and no formal process such as DELPHI was employed or required to establish consensus around its contents. To clarify this, we have added the following text on page 4 line 152 – “the SEEP-CI was developed by the authors with no formal input from external experts, thus no formal procedure for reaching consensus about the steps involved (e.g. the DELPHI method [ref added]) was conducted or required, and the steps of the SEEP-CI, described below, reflect solely the views of the authors.”

We accept that a limitation of our work is that it only reflects our proposed solution to the challenges of economic evaluation of complex health system interventions. The SEEP-CI would have stronger credibility if it reflected the views of a range of experts and those views had been brought in agreement using a formal research methodology, as the reviewer expected. To acknowledge this limitation, we have added the following text to the discussion  (line 443)– “we accept there are limitations to the SEEP-CI currently that would need to be addressed before it can be credibly used.  We have yet to test the SEEP-CI by using it from the start of the study. Also, the SEEP-CI is our proposed solution to the challenges of evaluating health system interventions, rather than reflecting the input of multiple experts synthesized through a formal consensus process, as was involved in developing guidelines such as CHEERS [ref added]. Further research validating the SEEP-CI, exploring its feasibility, acceptability to decision-makers, and ability to generate credible and helpful results, is required. A formal task force would be particularly suitable to developing the SEEP-CI further”.

    Accordingly, the author should describe the research methods of adjusting example(REACHOUT) separately, and so,  "2.3 Description of the SEEP-CI" should be moved to the "Results".

Authors’ response – We accept that the development and description of the SEEP-CI can be seen as results, and have moved section 2.3 to the Results accordingly. We are not sure which research methods are being referred to here – we provide an overview of the REACHOUT intervention and associated challenges in section 2, and provide references which give further details of the REACHOUT study.

    Moreover, the author should mentioned the analytic methods or process for identyfing critical values from the prior research results of REACHOUT program (related to 3.4)

Authors’ response – we have amended section 3.4 to clarify that the only critical values taken directly from REACHOUT study results were the costs associated with the intervention itself (section 3.4 page 9, see quote below).

Results.

     The author should present the results of economic evaluation about REACHOUT as tables and figures (costs, outcomes, ICER, NMB mentioned in 3.4).

Authors’ response. For the reasons set out in section 2, the economic evaluation of REACHOUT was a costing study, and it was not possible to directly measure the effectiveness of the intervention or calculate ICERs or NMB. We have added the following text to section 3.4 to clarify this – “Given the challenges described above, we were unable to robustly estimate effects, cost-effectiveness ratios, or net monetary benefit for REACHOUT. However, we were able to conduct included a detailed, country-specific exercise in costing the interventions evaluated, which has been reported elsewhere [10]. The key parameter from that costing exercise was the cost per CHW of QI.”

Discussion.

     Even though there are might be imitations for conducting economic evaluation of complex health system interventions based on existing protocols and guidelines, the author should clearly show the superiority of proposed guideline by comparing every contents with existing guidelines.

Authors’ response – Given that the SEEP-CI is at an early stage of development, we would not argue that the superiority of the SEEP-CI over existing economic evaluation guidelines has been established. In fact, the reviewer’s insightful suggestion has led us to realise that the SEEP-CI might best be seen as extending, rather than replacing, existing guidance. We have added the following text to the discussion on this point (line 433 page 10) “ It (the SEEP-CI) is intended to enhance, rather than replace, existing guidelines for the conduct and reporting of economic evaluations, such as those provided by the ISPOR CHEERS Taskforce [27]. Such guidelines set out core principles for economic evaluation that apply equally to health system interventions as health technologies, such as clearly defining the intervention and the decision population, identifying an appropriate time horizon and discount rates, providing sources for costs, and so on. We expect the SEEP-CI to be a helpful enhancement to existing guidance…”

Reviewer 2 Report

Thank you for the opportunity to review this manuscript. It is written clearly and is of interest to health economists especially, but also of general interest to others involved in trials and evaluations of complex interventions. The walkthrough of how the framework could be applied post-hoc to the REACHOUT study was a useful inclusion.

Please find below a list of queries/ suggestions to address in the manuscript.

P1, line 34: Please spell out the acronym SDG

P2, lines: 47-48: I would consider the NICE guidance on public health to be a relevant framework to mention here (https://www.nice.org.uk/process/pmg4/chapter/introduction).

P3, line 137: As the authors note, the outcomes arising from complex interventions can be wide-ranging. It may be useful to insert ‘health’ into the section header as the text refers to health outcomes solely. I see no reason why the framework could not be adopted for evaluations that use a broader societal perspective; perhaps the authors could explicitly state in this section that the relevant outcomes to include in an economic evaluation will be dependent on the perspective adopted.

P5, lines 214-227: The authors suggest using a proximal primary outcome; however, the next stage requires knowledge of the minimum health gains that the proximal impact would generate, and it isn’t clear how the health gains are to be determined. Perhaps it is just the phrasing at the end of the last sentence in the example given (‘rather than neonatal outcomes’), and the authors could clarify that data on these intrinsic outcomes is still required, but can be from existing literature if it is not feasible to collect alongside the study. The next section discusses the inclusion of health gains data well.

Discussion: The choice between universal or targeted intervention is relevant for system level interventions, along with the impact on health inequity. Can SEEP-CI be used to explore the impact on health inequity by including considerations for subgroup analyses?    

Author Response

We thank the reviewer for their efforts and their helpful and insightful feedback. We have considered their comments carefully and made amendments to our manuscript where appropriate. We give details of our responses and amendments below, in italics.

Thank you for the opportunity to review this manuscript. It is written clearly and is of interest to health economists especially, but also of general interest to others involved in trials and evaluations of complex interventions. The walkthrough of how the framework could be applied post-hoc to the REACHOUT study was a useful inclusion.

Authors’ response – we thank the reviewer for their kind words on our work, and their helpful feedback. We give our responses to each point below.

Please find below a list of queries/ suggestions to address in the manuscript.

P1, line 34: Please spell out the acronym SDG

Authors’ response - Done

P2, lines: 47-48: I would consider the NICE guidance on public health to be a relevant framework to mention here (https://www.nice.org.uk/process/pmg4/chapter/introduction).

Authors response – We agree and have added it to our list of relevant guidance.

P3, line 137: As the authors note, the outcomes arising from complex interventions can be wide-ranging. It may be useful to insert ‘health’ into the section header as the text refers to health outcomes solely. I see no reason why the framework could not be adopted for evaluations that use a broader societal perspective; perhaps the authors could explicitly state in this section that the relevant outcomes to include in an economic evaluation will be dependent on the perspective adopted.

Authors response – We completely agree that outcomes beyond health may well be relevant when evaluating a complex intervention. In our view, there is no reason why such outcomes cannot be incorporated in our framework. As the reviewer suggests, the decision perspective will help determine what outcomes are relevant. For this reason, we would prefer not to suggest in the section header that we are only considering health outcomes, but we are happy to add the following to the end of the section (line 187, page 5) “Also, the relevance of an outcome will depend on the decision perspective. Our example are health related, but outcomes beyond health should also be considered at this step if they are likely to be of importance to the decision-maker.”

P5, lines 214-227: The authors suggest using a proximal primary outcome; however, the next stage requires knowledge of the minimum health gains that the proximal impact would generate, and it isn’t clear how the health gains are to be determined. Perhaps it is just the phrasing at the end of the last sentence in the example given (‘rather than neonatal outcomes’), and the authors could clarify that data on these intrinsic outcomes is still required, but can be from existing literature if it is not feasible to collect alongside the study. The next section discusses the inclusion of health gains data well.

Authors’ response – The reviewer touches on a key issue at the heart of the SEEP-CI – the challenge of attributing impact to an intervention when the causal chain between them is complex. Our response is to ensure that the study is designed to test whether or not the intervention does impact the outcomes closest to it in the causal chain, even if they are instrumental. The reviewer’s point that this does not preclude collecting data on intrinsic outcomes as well is well taken. However, we expect that it will often be challenging to derive robust estimates of effect size, for all the reasons set out in section 2 of our paper. In this case, we recommend the use of decision modelling, informed by the literature (and expert opinion if required) in the next step. In order to clarify this, acknowledge the value of collecting data on distal as well as proximal outcomes, and strengthen the link between step 5 and step 6, we have added the following at the end of step 5 (2.3.5, page 7, line 267)

“ This does not preclude collecting data on a distal intrinsic outcome such as neonatal health, but it protects against the considerable risk of the study failing to show a significant attributable impact of the intervention on the distal intrinsic outcome, for all the reasons explored in section two. In this case, it should still be possible to establish that the intervention has had an impact, albeit on the proximal instrumental outcome. The resulting impact on the distal outcome of interest can then be inferred through analytical approaches than combine study results, the literature, and expert opinion, as per the next step”.

Discussion: The choice between universal or targeted intervention is relevant for system level interventions, along with the impact on health inequity. Can SEEP-CI be used to explore the impact on health inequity by including considerations for subgroup analyses?  

Authors’ response – This is an interesting question. In setting out the SEEP-CI, we have focused on allocative efficiency. However, it is clear that decision-making will usually draw on broader objectives such as access and health equity, and it would be important to reflect that in any guidance for economic evaluation of health system interventions. This would be interesting to explore in our next steps with the SEEP-CI, and we have added this to the discussion (line 463 page 11) “Furthermore, while the approach we set out here focuses on allocative efficiency, decision-makers are often interested in addressing health inequity as well as maximizing health gains. Further development of the SEEP-CI could involve extending the approach to reflect this need, for example by including subgroup analyses, or ensuring equity-based outcomes are formally included.”